# Adrenal suppression in patients with chronic obstructive pulmonary disease treated with glucocorticoids: Role of specific glucocorticoid receptor polymorphisms

**Pradeesh Sivapalan**[1,2]*, **Stina Willemoes Borresen**[3], **Josefin Eklöf**[1], **Marianne Klose**[3], **Freja S. Holm**[1], **Ulla Feldt-Rasmussen**[3,4], **Maria Rossing**[5], **Niklas R. Jørgensen**[4,6], **Rasmus L. Marvig**[5], **Mohamad Isam Saeed**[1], **Torgny Wilcke**[1], **Niels Seersholm**[1], **Alexander G. Mathioudakis**[7,8], **Jørgen Vestbo**[7,8], **Jens-Ulrik Stæhr Jensen**[1,4]

1 Section of Respiratory Medicine, Department of Medicine, Herlev and Gentofte Hospital, University of Copenhagen, Hellerup, Denmark, 2 Department of Internal Medicine, Zealand Hospital, University of Copenhagen, Roskilde, Denmark, 3 Department of Medical Endocrinology and Metabolism, Rigshospitalet, Copenhagen University Hospital, 4 Institute for Clinical Medicine, Faculty of Health Sciences, University of Copenhagen, Copenhagen, Denmark, 5 Center for Genomic Medicine, Rigshospitalet, Copenhagen University Hospital, Copenhagen, Denmark, 6 Department of Clinical Biochemistry, Copenhagen University Hospital Rigshospitalet, Copenhagen, Denmark, 7 Division of Infection, Immunity and Respiratory Medicine, University of Manchester, Manchester Academic Health Science Centre, Manchester, United Kingdom, 8 North West Lung Centre, Manchester University NHS Foundation Trust, Manchester, United Kingdom

* pradeesh.sivapalan.02@regionh.dk

**Data Availability Statement:** All relevant data are within the paper.

## Abstract

### Background

Single-nucleotide polymorphisms (SNPs) of the glucocorticoid receptor (GR) gene *NR3C1* have been associated with an altered sensitivity to glucocorticoids, and thus may alter the therapeutic effects of glucocorticoids. We investigated the prevalence of adrenal suppression after treatment with glucocorticoids and evaluated whether GR SNPs were associated with altered risks of adrenal suppression and metabolic disorders in patients with chronic obstructive pulmonary disease (COPD).

### Methods

In an observational prospective cohort study, we recruited 78 patients with severe COPD receiving 5 days glucocorticoid treatment for an exacerbation of COPD. In total, 55% of these patients were also receiving regular inhaled corticosteroids (ICS). Adrenal function was evaluated with a corticotropin test 30 days after the exacerbation. Patients were genotyped for *Bcl1*, *N363S*, *ER22/23EK*, and *9β* SNPs.

### Results

The prevalence of adrenal suppression (corticotropin-stimulated plasma-cortisol $\leq$ 420 nmol/L) 30 days after glucocorticoid treatment was 4/78 (5%). There was no difference between carriers and non-carriers of the polymorphisms (*Bcl1*, *9β*, *ER22/23K*, and *N363S*)

**Funding:** This study was funded by the Danish Regions Medical Fund (5894/16), the Danish Council for Independent Research (6110-00268B) and Herlev-Gentofte University hospital. The research salary of PS was sponsored by Herlev-Gentofte Hospital. The research salary of UFR was sponsored by an unrestricted research grant from the Novo Nordisk Fund. The research salary of SWB was sponsored by Skibsreder Per Henriksen, R. & Hustru's Foundation; Eva Madura's Foundation; and The Research Foundation of Copenhagen University Hospital, Rigshospitalet. Alexander G. Mathioudakis and Jørgen Vestbo are supported by the National Institute for Health Research (NIHR) Manchester Biomedical Research Centre (BRC). The funders did not play any role in the study design, data collection and analysis, decision to publish, or preparation of the manuscript.

**Competing interests:** The authors have declared that no competing interests exist.

**Abbreviations:** CAT, COPD assessment test; COPD, chronic obstructive pulmonary disease; GR, glucocorticoid receptor; HPA, hypothalamic–pituitary–adrenal; ICS, inhaled corticosteroids; SNP, single-nucleotide polymorphism.

in corticotropin stimulated plasma-cortisol concentrations. In the haplotype analyses, we included the 50 patients who had a high-sensitivity (76%), a low-sensitivity (4%), or a wild-type (20%) GR haplotype. There was no difference in the frequency of adrenal suppression or metabolic disorders between the two stratified groups: (a) high-sensitivity (*Bcl1* and/or *N363S*) haplotypes vs. (b) low-sensitivity (*9β* and/or *ER22/23K*) plus wild-type haplotypes ($p > 0.05$). Carriers of the high-sensitivity GR gene haplotype exhibited a steeper decline in stimulated P-cortisol with increased ICS dose (slope, $-1.35$ vs. $0.94$; $p = 0.17$), compared to the group with low-sensitivity or wild-type haplotypes, respectively.

## Conclusions

In total, 5% of patients exhibited insufficient adrenal function. The *Bcl1* and *N363S* polymorphisms did not seem to increase the risk of glucocorticoid suppression or metabolic disorders in adults treated with glucocorticoids for COPD exacerbations.

## 1. Introduction

Systemic glucocorticoid therapy (30–40 mg prednisolone) is commonly used to treat patients for acute exacerbation of chronic obstructive pulmonary disease (COPD), irrespective of disease etiology or phenotype [1]. This therapy has beneficial effects, such as shorter admission periods and more rapid improvements in pulmonary function. However, it is associated with several side effects [2, 3], including severe psychological and somatic side effects [4]. Short-term glucocorticoid therapy was shown to suppress the hypothalamic–pituitary–adrenal (HPA) axis in 45% to 63% of patients with a COPD exacerbation [5]. The degree of adrenal cortex suppression caused by glucocorticoid therapy varies considerably among individuals, and it is difficult to predict whether a particular patient will develop adrenal suppression or adrenal cortex failure [6, 7]. Previous studies found no correlation between glucocorticoid dose or therapy duration and the function of the HPA axis [8].

The glucocorticoid receptor (GR), encoded by the *NR3C1* gene, mediates the effects of glucocorticoids, and many polymorphisms in the GR gene have been associated with altered sensitivity to glucocorticoids [9, 10]. In patients with rheumatoid arthritis, four polymorphisms in the *NR3C1* gene were identified as clinically relevant. The polymorphisms *N363S* and *Bcl1* have been associated with a decreased risk of developing rheumatoid arthritis (i.e., are high-sensitivity GR gene single-nucleotide polymorphisms [SNPs] that protect against the autoimmune disease), and the polymorphisms *ER22/23EK* and *9β* are associated with an increased risk of developing rheumatoid arthritis [11].

Although discrepancies have been reported, the first two haplotypes appear to be predictors of obesity, dyslipidemia, and hypertension [10]. *Bcl1* is associated with a higher risk of hyperglycemia, increased insulin secretion, and abdominal obesity [12]. Furthermore, *N363S* has been associated with a higher risk of metabolic syndromes, type 2 diabetes mellitus, and cardiovascular disease [13], whereas *ER22/23EK* and *9β* were associated with more favorable metabolic profiles [10]. Notably, patients with steroid-resistant asthma are more likely to have the 9β polymorphism. However, the clinical significance of these polymorphisms for patients with COPD who undergo glucocorticoid therapy is unknown [14, 15].

To our knowledge, no study has investigated the association between GR haplotypes and adrenal suppression in patients with COPD following administration of glucocorticoids. We

assessed the prevalence of adrenal suppression after treatment with glucocorticoids and evaluated whether high glucocorticoid-sensitivity haplotypes were associated with an increased risk of adrenal suppression or metabolic disorders in patients with COPD.

## 2. Materials and methods

### 2.1. Study design and patients

This prospective population-based cohort study included 78 patients recruited 1 month after admission with a COPD exacerbation (47% males; median age, 75 years; range, 70–83 years) between 17 August 2017 and 1 February 2019 at the Respiratory Medicine department at Herlev and Gentofte University Hospital in Copenhagen, Denmark. All patients were asked if they would like to participate in the project during their admission for an exacerbation of COPD. Only white individuals were recruited because there may be genetic differences that affect the haplotypes in other populations.

Inclusion criteria were: (i) a diagnosis of COPD in white individuals aged 18 years or more; (ii) COPD exacerbation treated with glucocorticoids; and (iii) signed informed consent. Exclusion criteria were: (i) treatment with estrogen-containing medication, including contraceptives, less than 6 weeks before the corticotropin test; (ii) pregnancy or lactation; (iii) severe mental illness not adequately controlled by medication; (iv) detention by law for psychiatric treatment; and (vi) permanent systemic glucocorticoid therapy. Also, inhaled corticosteroid (ICS) treatment was paused 24 h before the corticotropin test to avoid cross-reactivity with cortisol in the assay. Patients were divided into an expected high-sensitivity GR gene haplotype group (patients carrying *Bcl1* and/or *N363S*, but not *ER22/23EK* and *9β*) and an expected low-sensitivity (patients carrying *ER22/23EK* and/or *9β*, but not *Bcl1* or *N363S*) plus wild-type (patients who exhibited wild-type genotypes for all four polymorphisms) GR gene haplotype group. Patients who had a mixture of high-sensitivity and low-sensitivity genotypes were excluded from the haplotype analyses.

The study was approved by The Regional Research Ethics Committee (VEK) with approval number H-15012207. All study procedures were carried out in accordance with the Declaration of Helsinki, and all participants provided written informed consent. The study was registered at clinicaltrials.gov (NCT03140761) before recruitment of patients.

### 2.2. Sequences of primers used in PCR

| *Polymorphism* | Forward | Reverse |
|---|---|---|
| *Bcl1* | TGCACCATGTTGACACCAATTCCT | GGTCTTGCTCACAGGGTTCTTGCC |
| *N363S* | TCATCGAACTCTGCACCCCTGG | AGTTGTCATCTCCAGATCCTTGGCA |
| *9β* | CCTACGCAGTGAAATGTCAGACTGT | TGCCAATTCGGTACAAATGTGTGGT |
| *ER22/23EK* | AACCCCAGCAGTGTGCTTGC | AGCGACAGCCAGTGAGGGTGA |

### 2.3. Exposure to ICS

All ICS prescriptions (alone or in a combination inhaler) prior to cohort entry were identified. These included: beclomethasone, budesonide, fluticasone, ciclesonide, and mometasone. All doses of ICS were converted to budesonide-equivalent doses: Beclomethasone and mometasone were considered equivalent to budesonide. Fluticasone propionate and ciclesonide were converted to budesonide doses using ratios of 2:1 and 2.5:1, respectively.

## 2.4. Procedures

All enrolled study patients were invited for a site visit 1 month after the acute exacerbation that included fasting venipuncture to measure fasting glucose, glycated hemoglobin levels, concentrations of the bone turnover markers C-terminal telopeptide of type 1 collagen and procollagen type 1 N-terminal propeptide, triglycerides, total cholesterol, as well as low-density and high-density lipoprotein cholesterol. DNA was extracted from peripheral venous blood samples using the QIAamp DNA mini kit (Qiagen, Hilden, Germany). DNA was genotyped for the four functional GR polymorphisms *Bcl1*, *9β*, *N363S*, and *ER22/23EK*. PCR reactions and genotyping procedures were carried out using the "allelic discrimination" technique, customized primers and probes, and the "Assay by Design service" provided by Applied Biosystems (Nieuwerkerk aan den IJssel, The Netherlands), in accordance with the manufacturer's instructions. Primers were designed using Primer-BLAST [16] and ordered through Eurofins [17]. Corticotropin tests were performed and blood samples collected 30 days after glucocorticoid treatment for COPD exacerbation was completed. For all patients, weight, height, body mass index (BMI), systolic and diastolic blood pressure, COPD assessment test (CAT) and waist and hip circumference were measured. CAT is a simple test to measure the patient-completed quality of life that contains eight questions covering the impact of symptoms in COPD.

## 2.5. Corticotropin tests

Adrenal cortisol secretion capacity was assessed using a short corticotropin test measuring the plasma cortisol concentration before and 30 min after intravenous administration of 250 μg corticotropin (Synacthen®; Atnahs Pharma, Basildon, UK). The threshold for diagnosing adrenal suppression using the short corticotropin test was based on adult reference concentrations (peak cortisol ≤ 420 nmol/L) from the same laboratory and population [18, 19].

## 2.6. Statistical analysis

The sample size calculation was designed for the haplotype analyses. We performed a *t*-test in which the ratio among the groups was 3:1 because previous results from other cohorts suggested that patients would exhibit this distribution of haplotypes [20]. The mean difference in cortisol concentration between patients with the wild-type plus *ER22/23EK* and/or *9β* polymorphisms and those with the homo/heterozygous *Bcl1* and or *N363S* polymorphisms was set to 100 nmol/L. These variables were based on data from two previous rheumatology studies [11, 21]. Power was set to 80% and the type 1 error rate was 5%. The test was single-sided, because we did not expect the most sensitive group to respond better to the corticotropin test. Standard deviation was set to 137 nmol/L. Consequently, 16 and 48 patients were included in each of the two groups (i.e., 64 patients in total). Assuming approximately 15–20% of the recruited patients would not be placed in either of the two groups, additional 11–16 patients were needed, resulting in a decided recruitment of 78 patients to the study. Continuous data were expressed as means with standard deviations when normally distributed or as medians with interquartile ranges when non-normally distributed. Categorical variables were expressed as counts and percentages. Comparisons among groups were made using *t*-tests or Mann–Whitney U tests for continuous variables and $\chi^2$ tests or Fisher's exact test when appropriate for categorical variables. For each genotype, the relationship between total systemic glucocorticoid dose over the preceding 6 months and adrenal suppression was evaluated using linear regression models. Spearman's correlation was used to evaluate the correlation between high- and low-sensitivity GR gene polymorphisms and stimulated cortisol levels. Statistical analyses were carried out using SAS software (ver. 9.4; SAS Institute, Inc., Cary, NC, USA) and a *p*-value < 0.05 was considered statistically significant.

## 3. Results

In total, 78 patients were included in the study. Baseline characteristics are given in Table 1. In total, 46% of patients had hypertension, 10% had ischemic heart disease, 6% had heart failure, 13% had diabetes, and 3% had kidney disease. Overall, 4/78 (5%) of patients had an insufficient response to the corticotropin test 1 month after treatment with corticosteroids with the most suppressed stimulated cortisol level of 280 nmol/L. Two of the 38 patients with a *Bcl1* or *N363S* genotype and none of the 12 patients with *ER22/23EK* or *9β* genotype had suppressed glucocorticoid function.

The SNP frequencies were: *N363S*, 3.9%; *Bcl1*, 63.2%; *ER22/23EK*, 2.6%; and *9β*, 18.4%. Corticotropin stimulated plasma cortisol was negatively associated with male sex, whereas no correlation was found with BMI, age, accumulated systemic glucocorticoid use 6 months before the test, or days since completing the course of systemic glucocorticoids after the COPD exacerbation (Table 2). For the haplotype analyses, we included the subgroup of 50 patients with the expected high-sensitivity (76%) or low-sensitivity (4%) GR gene haplotypes plus wild-type GR haplotype (20%). The baseline characteristics of the two groups were similar, except for ischemic heart disease, which was more prevalent in the low-sensitivity plus wild-type haplotype group (25%) than in the high-sensitivity haplotype group (3%; $p = 0.01$); (Table 1).

There was no correlation between accumulated systemic glucocorticoids and 30 min cortisol concentrations in the high-sensitivity haplotype group ($p$-value, 0.79) or the low-sensitivity plus wild-type haplotype group ($p$-value, 0.26) (Fig 1 and Table 3). When all 78 patients were analyzed, corticotropin stimulated plasma-cortisol levels did not differ between carriers and non-carriers of *Bcl1* ($p = 0.74$), *9-beta* ($p = 0.33$), *ER22/23K* ($p = 0.37$), or *N363S* ($p = 0.35$; Table 4).

Carriers of the high-sensitivity GR gene haplotype exhibited a steeper decline in stimulated P-cortisol with increased ICS dose compared to patients with the low-sensitivity haplotype plus those with wild-type haplotypes (slope, –1.35 vs. 0.94; $p = 0.17$; Fig 2 and Table 5).

There was no difference between corticotropin stimulated plasma cortisol or metabolic disorders in patients with the high-sensitivity haplotypes vs. low-sensitivity plus wild-type haplotypes ($p > 0.05$; Table 6).

## 4. Discussion

In this study, we found that 5% of patients exhibited insufficient glucocorticoid function 30 days after undergoing treatment for COPD exacerbation, and there was a clear association with concomitant ICS dose. Compared with the *ER22/23EK*, *9β* and wild-type haplotypes, the *Bcl1* and *N363S* haplotypes were not associated with an increased risk of adrenal suppression or metabolic disorders in patients with COPD exacerbation treated with glucocorticoids. In addition, there was no difference between carriers and non-carriers of any of the four gene polymorphisms in terms of stimulated cortisol concentrations. Here, we determined the prevalence of adrenal suppression 1 month after exacerbation in a cohort of patients with severe COPD, 49% of whom received ICS treatment. We would expect more patients to exhibit insufficiency immediately after exacerbation. Previous research has shown that several patients were insufficient soon after discontinuation of oral glucocorticoids, although many regained adrenal function in the weeks and months afterward; however, a few of these patients remained insufficient several years later or for the rest of their lives [22]. This patient group, with severe COPD, are at risk from repeated courses of glucocorticoid treatment, which may result in suppression of the HPA axis and the development of chronic adrenal insufficiency. Clinicians should be aware of this risk to ensure correct clinical management of such patients.

**Table 1. Clinical, functional, and biological characteristics of COPD patients with high-sensitivity haplotypes, those with low-sensitivity or wild-type haplotypes, and 'mixed type', i.e. patients with a mixture of high-sensitivity and low-sensitivity genotypes.** Mixed type was excluded from haplotype analyses.

| | All (n = 78) | High sensitivity: *Bcl1* or *N363S* (n = 38) | Low sensitivity: *ER22/23EK* and *9β* plus wild-type (n = 12) | *p*–value (High vs. Low sensitivity) | Mixed type (n = 28) |
|---|---|---|---|---|---|
| Age [years], median (IQR) | 75 (70 to 83) | 74 (68 to 81) | 75 (72 to 87) | | 77 (72 to 82.5) |
| Male (%) | 46.8 | 44.7 | 54.5 | | 46.4 |
| Current or former smoker (%) | 97.4 | 97.4 | 100 | | 96.4 |
| Never smoker (%) | 2.6 | 2.6 | 0 | | 3.6 |
| Pack years, median (IQR) | 45 (33 to 60) | 45 (32.5 to 60) | 48 (35 to 54) | | 41 (22 to 52.5) |
| Weekly alcohol consumption [units], median (IQR) | 2 (0 to 8) | 1 (0 to 7) | 1.5 (0 to 7) | | 6 (1.5 to 12) |
| Body mass index [kg/m$^2$] | 25.8 (24.4 to 27.2) | 25.6 (22.2 to 28.9) | 25.3 (23.3 to 27.3) | | 26.7 (24.2 to 29.1) |
| **GOLD Class** | | | | 0.91 | |
| Class A n (%) | 1 (1.35) | 1 (2.78) | 0 (0.00) | | 0 (0.00) |
| Class B n (%) | 11 (14.86) | 5 (13.89) | 2 (18.18) | | 4 (14.29)?? |
| Class C n (%) | 8 (10.81) | 5 (13.89) | 2 (18.18) | | 1 (3.57) |
| Class D n (%) | 54 (72.97) | 25 (69.44) | 7 (63.64) | | 22 (78.57) |
| **Medication** | | | | | |
| Regular ICS (%) | 55 | 58 | 58 | 0.98 | 39 |
| Regular ICS daily dose [μg] | 1030.9 (819 to 1243) | 884 (609 to 1158) | 1257 (475 to 2040) | 0.46 | 652 (476 to 828) |
| Days after last completed course of corticosteroids, median (IQR) | 28 (25 to 31) | 28 (22 to 32) | 28 (26 to 31) | 0.49 | 28 (24.5 to 30) |
| Accumulated systemic GC use 6 months before corticotropin test [mg] | 324 (289 to 358) | 356 (252 to 461) | 264 (199 to 328) | 0.85 | 367 (302 to 432) |
| LAMA (%) | 83.3 | 76.3 | 91.7 | 0.25 | 89.3 |
| LABA (%) | 80.8 | 79.0 | 83.3 | 0.74 | 82.1 |
| **Comorbidity** | | | | | |
| Hypertension (%) | 46.2 | 42.1 | 58.3 | 0.33 | 46.4 |
| Atrial fibrillation (%) | 12.8 | 13.2 | 16.7 | 0.76 | 10.7 |
| Heart failure (%) | 6.4 | 8.3 | 7.9 | 0.96 | 3.6 |
| Kidney failure (%) | 2.6 | 5.3 | 0 | 0.42 | 0 |
| Diabetes mellitus (%) | 12.8 | 18.4 | 8.3 | 0.41 | 7.1 |
| Ischemic heart disease (%) | 10.3 | 2.6 | 25 | 0.01 | 14.3 |
| Asthma (%) | 10.3 | 13.2 | 0 | 0.19 | 10.7 |
| **Paraclinical parameters** | | | | | |
| Hemoglobin [mmol/L] | 8.9 (8.2 to 9.6) | 9.2 (8.4 to 10.0) | 8.9 (8.4 to 9.4) | | 8.6 (8.3 to 8.9) |
| Leucocytes [10E9/L] | 7.5 (6.4 to 9.4) | 8.0 (7.3 to 8.8) | 8.2 (7.1 to 9.4) | | 8.3 (7.3 to 9.4) |
| Thrombocytes [10E9/L] | 300 (245 to 374) | 303 (270 to 336) | 296 (245 to 347) | | 329 (291 to 366) |
| Albumin [g/L] | 42.2 (41.3 to 43.0) | 42.8 (41.6 to 44.1) | 41.4 (38.4 to 44.3) | | 41.6 (40.7 to 42.6) |
| Potassium [mmol/L] | 3.9 (3.8 to 4.0) | 3.9 (3.8 to 4.0) | 4.0 (3.7 to 4.2) | | 3.9 (3.8 to 4.1) |
| Sodium [mmol/L] | 140.8 (140.2 to 141.4) | 141.2 (140.3 to 142.1) | 139.9 (139.2 to 140.6) | | 140.7 (139.6 to 141.8) |
| Carbamide [mmol/L] | 6.2 (5.6 to 6.7) | 5.8 (5.1 to 6.4) | 7.4 (5.3 to 9.6) | | 6.2 (5.4 to 7.0) |
| Creatinine [μmol/L] | 79.9 (75.2 to 84.6) | 78.1 (71.1 to 85.0) | 86.8 (75.6 to 98.0) | | 79.7 (71.5 to 87.8) |
| eGFR [mL/min] | 71.8 (68.4 to 75.3) | 73.5 (68.4 to 78.6) | 66.1 (55.1 to 77.1) | | 71.8 (66.6 to 77.0) |

(*Continued*)

**Table 1.** (Continued)

|  | All (*n* = 78) | High sensitivity: *Bcl1* or *N363S* (*n* = 38) | Low sensitivity: *ER22/23EK* and *9β* plus wild-type (*n* = 12) | *p*–value (High vs. Low sensitivity) | Mixed type (n = 28) |
|---|---|---|---|---|---|
| ASAT [U/L] | 47.4 (22.3 to 72.5) | 34.7 (29.0 to 40.4) | 34.5 (25.7 to 43.4) |  | 35.3 (32.7 to 37.9) |
| ALAT [U/L] | 27.4 (25.3 to 29.4) | 26.9 (24.1 to 29.7) | 25.2 (17.8 to 32.5) |  | 28.8 (25.4 to 32.2) |
| Basic phosphatase [U/L] | 75.0 (70.1 to 79.2) | 75.4 (69.1 to 81.7) | 72.3 (58.5 to 86.1) |  | 75.6 (69.2 to 82.0) |
| INR | 1.0 (0.9 to 1.1) | 1.2 (1.0 to 1.3) | 1.1 (0.8 to 1.5) |  | 1.0 (0.9 to 1.0) |
| Fasting blood glucose [mmol/L] | 6.0 (5.8 to 6.3) | 6.2 (5.7 to 6.6) | 5.8 (5.4 to 6.2) |  | 6.0 (5.6 to 6.3) |
| Corticotropin [pmol/L] | 4.8 (4.0 to 5.5) | 4.9 (3.9 to 5.9) | 4.3 (1.5 to 7.1) |  | 4.6 (3.2 to 6.0) |
| Vitamin D [nmol/L] | 77.2 (70.7 to 83.6) | 78.9 (70.3 to 87.5) | 80.4 (54.6 to 106.2) |  | 73.6 (63.1 to 84.2) |
| PTH [pmol/L] | 7.3 (6.5 to 8.1) | 6.7 (5.6 to 7.8) | 9.6 (7.4 to 11.8) |  | 7.2 (6.0 to 8.3) |
| TSH [U/L] | 1.7 (1.3 to 2.0) | 1.7 (1.3 to 2.1) | 1.9 (0.18 to 3.6) |  | 1.6 (1.1 to 2.0) |

Data are expressed as means (95% confidence interval) unless otherwise stated. Abbreviations: GC, glucocorticoids; GOLD, Global initiative for chronic Obstructive Lung Disease; LABA, long-acting beta-agonist; LAMA, long-acting muscarinic antagonist; ICS, inhaled corticosteroids; COPD, chronic obstructive pulmonary disease; IQR, interquartile range; eGFR, estimated glomerular filtration rate; PTH, Parathyroid hormone; TSH, thyroid-stimulating hormone; ALAT, alanine aminotransferase; ASAT, aspartate transaminase; INR, international normalized ratio.

We did find a correlation between daily ICS dose and adrenal suppression (Fig 2) in the high-sensitivity GR gene haplotype group, although a higher proportion of patients in the low-sensitivity plus wild-type GR gene haplotype group received ICS treatment. Similarly, previous studies found that different genotypes were associated with ICS-induced adrenal suppression in children with asthma [23–25]. One of these studies, a genome-wide association study (*n* = 407), found that a common variant in the PDGFD locus was associated with an increased risk of adrenal suppression. Unlike our study, these analyses were performed in a cohort of children with less severe disease, and patients were presumably treated with fewer glucocorticoid rescue courses. Other genetic variants are reportedly relevant for glucocorticoid sensitivity, but currently none of these have been introduced as biomarker in routine clinical practice. Biomarkers predicting the risk of adrenal insufficiency might enable clinicians to inform patients about the level of risk and to train patients to recognize factors that may increase this risk level (such as infection, trauma, or surgery). Increasing awareness of the importance of glucocorticoid supplements in these situations would also reduce the risk of adrenal crisis.

The effects of glucocorticoids on tissues are influenced by glucocorticoid sensitivity, which may depend partly on functional SNPs in the GR gene. Dexamethasone suppression testing

**Table 2. Relationship between 30 min cortisol response and possible confounders in the total cohort (N = 78).**

|  | Spearman's Rho | *p*-value |
|---|---|---|
| Male | −0.24 | 0.04 |
| Body mass index | −0.15 | 0.22 |
| Age | 0.06 | 0.63 |
| Accumulated GC use 6 months before test | −0.14 | 0.23 |
| Days since completing the course of glucocorticoids | 0.09 | 0.47 |

Abbreviations: GC, glucocorticoids.

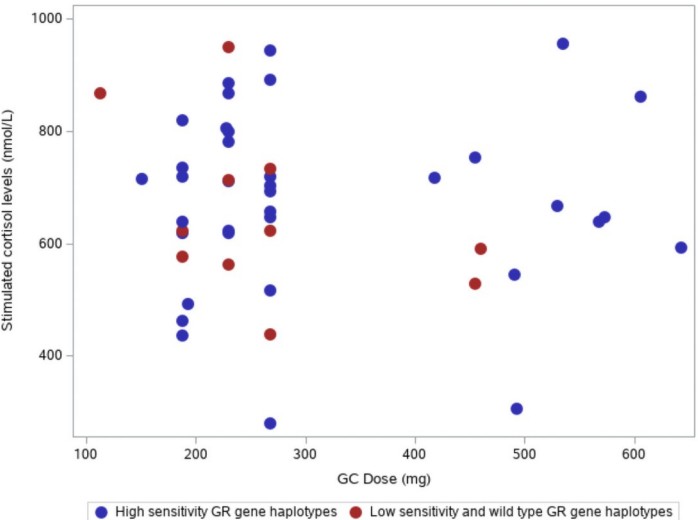

**Fig 1. The relationship between total accumulated systemic glucocorticoid dose (prednisolone equivalent dose, mg) in the 6 months before the corticotropin test and stimulated cortisol concentrations for the high-sensitivity glucocorticoid receptor gene haplotype group (*p*-value 0.79) compared with the low-sensitivity and wild-type glucocorticoid receptor gene haplotype group (*p*-value 0.26).** Abbreviations: GC, glucocorticoids; GR, glucocorticoid receptor.

has shown that the *Bcl1* haplotype increases glucocorticoid sensitivity *in vivo* [26], and this is correlated with increasing BMI and central adiposity, as well as insulin resistance [27]. Several SNPs in the GR gene influence sensitivity to glucocorticoids and have been linked with metabolic syndromes. However, the data include discrepancies, perhaps due to heterogeneity among the studied populations and the limited number of samples [13]. Glucocorticoids are widely used to treat a variety of lung diseases including COPD, asthma and interstitial lung diseases, and the effects of glucocorticoid treatment vary considerably among patients. Some patients appear to respond well to glucocorticoid therapy but also develop serious side effects. In contrast, other patients require very high glucocorticoid doses to achieve clinical effects and do not exhibit side effects [28].

Previous studies have indicated a relationship between altered glucocorticoid sensitivity mediated by the *Bcl1* and *ER22/23EK* polymorphisms of the GR gene and changes in body composition and metabolism in healthy subjects [26, 28]. However, in our study, we found no apparent difference between subjects with *Bcl1* or *N363S* polymorphisms and those with *ER22/23EK* or *9β* polymorphisms in terms of metabolic disorders. This may be due to

**Table 3. The relationship between total systemic glucocorticoid dose (prednisolone equivalent dose) over the preceding 6 months and 30 min cortisol response was evaluated using Spearman's correlation.**

|  | Spearman's Rho | 95% CI | *p*-value |
|---|---|---|---|
| Low-sensitivity plus wild-type GR gene haplotypes (*n* = 12) | −0.37 | (−0.79 to 0.29) | 0.26 |
| High-sensitivity GR gene haplotype (*n* = 38) | −0.05 | (−0.36 to 0.28) | 0.79 |
| Pooled (*n* = 50) | −0.10 | (−0.37 to 0.19) | 0.49 |
| Mixed Type (*n* = 28) | −0.10 | (−0.47 to 0.30) | 0.64 |
| Total Cohort (*n* = 78) | −0.14 | (−0.36 to 0.09) | 0.23 |

Abbreviations: CI, confidence interval; GR, Glucocorticoid receptor.

**Table 4. Differences in stimulated plasma-cortisol for non-carriers vs. carriers of each polymorphism.**

| | Stimulated cortisol | p-value | Pearson's Rho |
|---|---|---|---|
| | Mean (95% CI) | | |
| **Bcl1** | | | |
| Non-carrier | 654 (606–702) | 0.74 | 0.04 |
| Carrier | 666 (609–724) | | |
| **9β** | | | |
| Non-carrier | 674 (627–721) | 0.33 | −0.12 |
| Carrier | 637 (577–697) | | |
| **ER22/23EK** | | | |
| Non-carrier | 663 (626–700) | 0.37 | −0.11 |
| Carrier | 561 (not calculated, n = 2) | | |
| **N363S** | | | |
| Non-carrier | 651 (612–689) | 0.35 | 0.11 |
| Carrier | 708 (617–799) | | |

Abbreviations: CI, confidence interval.

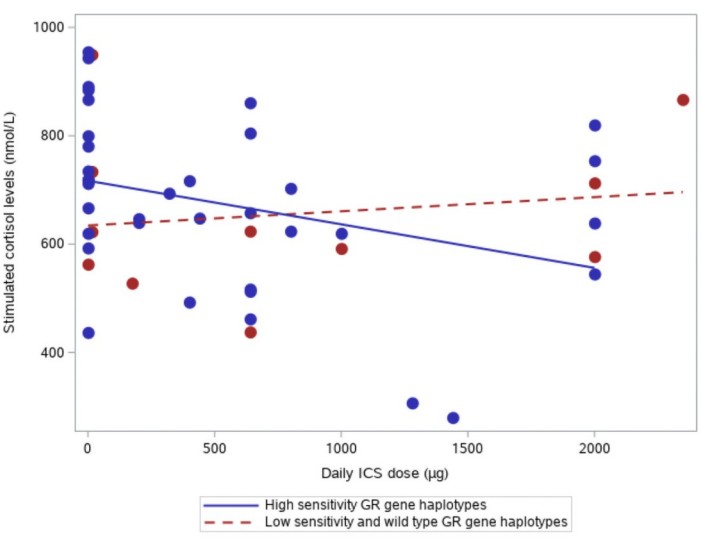

**Fig 2. The relationship between ICS dose 1 day before the Corticotropin test (µg) and stimulated cortisol concentrations at baseline for the high-sensitivity glucocorticoid receptor gene haplotype group compared with the low-sensitivity and wild-type glucocorticoid receptor gene haplotype group.** Abbreviations: ICS, inhaled corticosteroid; GR, glucocorticoid receptor.

differences in subject selection (i.e., healthy volunteers vs. patients with COPD), methodology, the small sample size and risk of type 2 error. Another study found a correlation between the *Bcl1* polymorphism and central adiposity, impaired glucose tolerance, and dyslipidemia in patients with Addison's disease [29]. However, these results were obtained from a more heterogeneous study population than ours (i.e., including differences in disease duration, glucocorticoid type, and dose).

Most of the effects of glucocorticoids are probably mediated by the GR. However, the response to glucocorticoids varies considerably among individual subjects, as shown by the variation in suppression in response to 0.25 mg of dexamethasone. Several polymorphisms in

**Table 5. The relationship between daily ICS dose and adrenal suppression assessed by linear regression.**

|  | Slope | 95% CI | *P*-value |
|---|---|---|---|
| Low-sensitivity plus wild-type GR gene haplotypes (*n* = 12) | 0.94 | (−3.59 to 5.46) | 0.65 |
| High-sensitivity GR gene haplotype (*n* = 38) | −1.35 | (−2.67 to −0.02) | 0.05 |
| Interaction |  |  | 0.43 |
| Pooled (*n* = 50) | −0.93 | (−2.27 to 0.41) | 0.17 |
| Mixed Type (*n* = 28) | 0.66 | (−1.40 to 2.72) | 0.51 |
| Total Cohort (*n* = 78) | −0.36 | (−1.46 to 0.74) | 0.51 |

Abbreviations: CI, confidence interval; GR, Glucocorticoid receptor.

the gene encoding the GR have been described. However, it is unclear to what extent the observed variations in response are due to GR polymorphisms or other factors.

A major strength of this study was that for the first time, we were able to study the relationship between GR haplotypes and adrenal suppression in patients with COPD 1 month after acute exacerbations were treated with oral glucocorticoids in addition to standard treatment. This relationship is important because many patients receive frequent treatments with glucocorticoids. Moreover, our homogeneous patient population also exhibited many comorbidities. Patients undergoing permanent systemic glucocorticoid therapy were not included because we were primarily interested in investigating patients who received a short course of glucocorticoids for a COPD exacerbation. All the cortisol analyses were performed in the same laboratory and with the same method as the earlier study [19].

**Table 6. Cortisol and Metabolic outcomes in patients with high-sensitivity haplotypes vs. low-sensitivity plus wild-type haplotypes.**

| Outcome | Bcl1 or N363S (*n* = 38) High-sensitivity GR gene haplotypes | ER22/23EK and 9β (*n* = 12) Low-sensitivity plus wild-type GR gene haplotypes | Unadjusted *p*-value* | Pearson's R |
|---|---|---|---|---|
| Basal cortisol [nmol/L] | 362 (325 to 398) | 307 (256 to 357) | 0.12 | 0.22 |
| Stimulated cortisol [nmol/L] | 675 (622 to 728) | 655 (555 to 756) | 0.71 | 0.06 |
| Adrenal suppression n (%) | 2 (5.4) | 0 | 0.43 | −0.11 |
| HbA1c [mmol/mol] | 39.8 (37.4 to 42.3) | 40.0 (37.9 to 42.1) | 0.94 | −0.01 |
| P1NP [µg/L] | 39.5 (32.7 to 46.2) | 41.8 (28.8 to 54.8) | 0.73 | −0.05 |
| CTX [ng/L] | 194.4 (144.4 to 244.3) | 247.4 (141.9 to 352.9) | 0.31 | −0.05 |
| Systolic blood pressure [mm Hg] | 137 (130 to 144) | 141 (127 to 155) | 0.62 | −0.07 |
| Diastolic blood pressure [mm Hg] | 78 (74 to 82) | 77 (70 to 85) | 0.82 | 0.03 |
| Pulse [heart rate/min] | 84 (78 to 90) | 80 (71 to 88) | 0.44 | 0.11 |
| Waist measurement [cm] | 97.6 (91.6 to 103.7) | 99.1 (90.3 to 107.9) | 0.81 | −0.04 |
| Hip measurement [cm] | 102.8 (98.5 to 107) | 102.3 (95.1 to 109.5) | 0.92 | 0.02 |
| HDL cholesterol [mmol/L] | 1.6 (1.4 to 1.8) | 1.3 (1.1 to 1.6) | 0.15 | 0.21 |
| LDL cholesterol [mmol/L] | 2.7 (2.4 to 3.1) | 2.6 (2.2 to 3.0) | 0.73 | 0.05 |
| VLDL cholesterol [mmol/L] | 0.7 (0.6 to 0.9) | 0.7 (0.5 to 0.9) | 0.88 | 0.02 |
| Triglycerides [mmol/L] | 1.6 (1.3 to 1.9) | 1.6 (1.2 to 1.9) | 0.83 | 0.03 |
| CAT score | 20.2 | 19.3 | 0.73 | 0.05 |

Data are expressed as medians (interquartile range) unless otherwise stated. Abbreviations: HbA1c, glycated hemoglobin; P1NP, procollagen type I N-terminal propeptide; CTX, C-terminal telopeptide of type 1 collagen; CAT, COPD assessment test; HDL, high-density lipoprotein; LDL, low-density lipoprotein; VLDL, very-low-density lipoprotein. *We planned to apply the Bonferroni correction. However, none of the p-values were significant. Therefore, we did not adjust for multiple testing.

Furthermore, our study population may be considered representative of a larger population because we included patients with COPD but had few exclusion criteria. A major limitation of our study was that few patients could be included in the comparison analysis. In small study populations like ours, there is a greater risk that any relationships observed are coincidental, and there is also a greater risk of overlooking genuine relationships. However, significant associations between genetic polymorphisms and severe adverse drug reactions have previously been identified from small cohorts and led to changes in clinical practice [30]. It has previously been shown that GR is involved in iatrogenic suppression of the HPA axis [31, 32]. The most common causes of adrenal insufficiency are otherwise pituitary tumors, adrenal hemorrhage, infections, and autoimmune disease [33]. To our knowledge, none of our patients had any of these conditions.

Unfortunately, we were unable to obtain the required number of patients with prespecified haplotypes that was suggested by the power calculation. Combining low-sensitivity GR gene haplotypes with wild-type haplotypes is a limitation of our study. Had we only included low-sensitivity GR gene haplotypes, we would have required many more patients. We decided that it was unnecessary to include more patients to test our hypothesis. We acknowledge that having low-sensitivity and high-sensitivity GR gene haplotype groups and excluding patients with wild-type haplotypes would have been optimal. Also, it would have been interesting to consider the effect of the different haplotypes present in the mixed genotype and wild type group, but this was not possible with the obtained study sample size. However, separate analyses of each of the SNPs from all 78 patients also found no correlation. Therefore, it is unlikely that a large effect has been overlooked. Nonetheless, our results will need to be validated by larger studies in the future. In addition, our cohort included many older patients with substantial comorbidities that may mask the effects of GR polymorphisms. Further studies, involving *ex vivo* models and reproduced alleles on plasmids, may validate the study rationale.

## 5. Conclusions

Our findings suggest that the *Bcl1* and *N363S* gene polymorphisms did not increase the risk of acute adrenal suppression in adults undergoing treatment with systemic glucocorticoids for COPD exacerbations. However, larger studies are needed to confirm this conclusion. Perhaps future studies applying whole-genome sequencing will identify other polymorphisms that may influence responses to glucocorticoids, including potential side effects.

## Acknowledgments

We are grateful to the Steering Committee of COP:TRIN (coptrin.dk) for providing input to the study during meetings.

## Author Contributions

**Conceptualization:** Pradeesh Sivapalan, Stina Willemoes Borresen, Marianne Klose, Ulla Feldt-Rasmussen, Maria Rossing, Niklas R. Jørgensen, Torgny Wilcke, Niels Seersholm, Jørgen Vestbo, Jens-Ulrik Stæhr Jensen.

**Data curation:** Pradeesh Sivapalan, Stina Willemoes Borresen, Marianne Klose, Freja S. Holm, Ulla Feldt-Rasmussen, Maria Rossing, Niklas R. Jørgensen, Rasmus L. Marvig, Niels Seersholm, Jørgen Vestbo, Jens-Ulrik Stæhr Jensen.

**Formal analysis:** Pradeesh Sivapalan, Maria Rossing, Rasmus L. Marvig, Jens-Ulrik Stæhr Jensen.

**Funding acquisition:** Pradeesh Sivapalan.

**Investigation:** Pradeesh Sivapalan, Stina Willemoes Borresen, Josefin Eklöf, Freja S. Holm.

**Methodology:** Pradeesh Sivapalan, Stina Willemoes Borresen, Josefin Eklöf, Marianne Klose, Ulla Feldt-Rasmussen, Rasmus L. Marvig, Jens-Ulrik Stæhr Jensen.

**Project administration:** Pradeesh Sivapalan, Jens-Ulrik Stæhr Jensen.

**Resources:** Pradeesh Sivapalan, Jens-Ulrik Stæhr Jensen.

**Software:** Pradeesh Sivapalan, Jens-Ulrik Stæhr Jensen.

**Supervision:** Pradeesh Sivapalan, Jens-Ulrik Stæhr Jensen.

**Validation:** Pradeesh Sivapalan, Jens-Ulrik Stæhr Jensen.

**Visualization:** Pradeesh Sivapalan.

**Writing – original draft:** Pradeesh Sivapalan, Jens-Ulrik Stæhr Jensen.

**Writing – review & editing:** Pradeesh Sivapalan, Stina Willemoes Borresen, Josefin Eklöf, Marianne Klose, Freja S. Holm, Ulla Feldt-Rasmussen, Maria Rossing, Niklas R. Jørgensen, Rasmus L. Marvig, Mohamad Isam Saeed, Torgny Wilcke, Niels Seersholm, Alexander G. Mathioudakis, Jørgen Vestbo, Jens-Ulrik Stæhr Jensen.

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
