## [Decision Letter · Decision Letter 0]

24 Jun 2021

PONE-D-21-06570

Adrenal suppression in patients with chronic obstructive pulmonary disease treated with glucocorticoids: role of specific glucocorticoid receptor polymorphisms

PLOS ONE

Dear Dr. Pradeesh Sivapalan,

Thank you for submitting your manuscript to PLOS ONE. After careful consideration, we feel that it has merit but does not fully meet PLOS ONE’s publication criteria as it currently stands. Therefore, we invite you to submit a revised version of the manuscript that addresses the points raised during the review process.

The manscript is interesting but it needs appropriate revision following reviewers' suggestions in order to reach the standard of the journal.

We look forward to receiving your revised manuscript.

Kind regards,

Fabio Luigi Massimo Ricciardolo

Academic Editor

PLOS ONE

Journal Requirements:

Additional Editor Comments (if provided):

The manucript is interesting but the authors should improve it following the reviewers' suggestions.

Reviewers' comments:

Reviewer's Responses to Questions

**Comments to the Author**

1. Is the manuscript technically sound, and do the data support the conclusions?

Reviewer #1: Yes

Reviewer #2: Partly

2. Has the statistical analysis been performed appropriately and rigorously? 

Reviewer #1: Yes

Reviewer #2: Yes

3. Have the authors made all data underlying the findings in their manuscript fully available?

Reviewer #1: Yes

Reviewer #2: Yes

4. Is the manuscript presented in an intelligible fashion and written in standard English?

Reviewer #1: Yes

Reviewer #2: No

5. Review Comments to the Author

Reviewer #1: Manuscript number: PONE-D- 21-06570

Adrenal suppression in patients with chronic obstructive pulmonary disease treated with glucocorticoids: role of specific glucocorticoid receptor polymorphisms

Sivapalan and co-workers investigated the possible association between single-nucleotide polymorphisms (SNPs) of glucocorticoid receptor (GR) and metabolic disorders in patients with chronic obstructive pulmonary disease (COPD). In particular, the authors focused on the connection between GR haplotypes and adrenal suppression in patients with COPD following administration of systemic glucocorticoid (therapy for COPD exacerbation treatment). The authors conclude that COPD patients treated with systemic GC for their exacerbation and characterized by BclI and N363S gene polymorphisms did not have a major risk of acute adrenal suppression or metabolic disorders.

It is an interesting study, but the following should be addressed:

Major comments:

1) The author in the Material and Methods section not reported the primer sequence used for genotyping the four GR polymorphisms. Please provide this information; it is useful for the readers that want to perform a similar analysis.

2) Please provide a new table with the clinical, functional and biologic characteristics of all 77 patients included in the study: i.e. the prevalence of male and female, COPD step based on GOLD guidelines, age, BMI, etc.

3) on Results section authors affirm that: “the ischaemic heart disease, which seemed more prevalent in the low plus wild type (25%) than 139 in the high (3%) GC-sensitivity group.” In my opinion, it is necessary to report this statistical significance in Table 1. This could be misleading for the readers.

4) Page 4 lines 131-134 authors underline that: “We found a negative association between male gender and stimulated cortisol response ( Rho = -0.24, p = 0.042), but no correlation to other possible confounders: BMI (Rho = -0.15, p =133 0.22), age (Rho = 0.06, p = 0.63), accumulated GC use 6 months before test (Rho = -0.14, p = 0.23) and days since completed GCs (Rho = 0.09, p = 0.47)”. Please provide a table or figure that show these findings.

Minor comments

1) Page 2, line 65: provide reference.

2) The authors did not indicate the inhaled corticosteroid considered for the analyses (beclomethasone or fluticasone). Please provide this information.

3) Page 3, line 97: replace the square with the symbol β.

4) Page 8, lines 225-227: consider moving the sentence in the “Corticotropin test” (Materials and Methods section).

Reviewer #2: In the present study, Sivapalan et al evaluate the role of glucocorticoid receptor polymorphisms in the development of adrenal suppression following ICS treatment for COPD exacerbations. As rightly pointed out in the discussion, uniting subjects with low-sensitive haplotypes with mixed ones (wild-type) does not make particular sense, if not to increase the numerosity. In this way, however, the real comparison between low-sensitive and high-sensitive is lost. In fact, I would expect wild-types to have an intermediate behavior between low- and high-sensitive haplotypes. How many subjects are actually with low-sensitive haplotypes and how many are wild-type? Furthermore:

- The authors enroll 77 patients, but the description in table 1 is only about 50. What characteristics do these additional 27 have and why were they excluded?

- What is the real distribution of haplotype sets in the population? The individual SNPs are instead reported.

- In estimating the sample size, how do the authors define the reported corticotropin values and why do they choose a 3:1 ratio?

- In the materials and methods it is not clear how the continuous data are expressed.

- In Table 3 and Figure 2, the authors found an inverse association between ICS and P-cortisol only in high-sensitive subjects (which are 2.5 times more numerous than low + wild-types).

- Figure 1 shows a value probably to be considered an outlier.

- Tables are not numbered consecutively

- There are typos

The work is interesting, however, it should be re-evaluated with a better subdivision of the groups.

6. PLOS authors have the option to publish the peer review history of their article (what does this mean?). If published, this will include your full peer review and any attached files.

Reviewer #1: No

Reviewer #2: No

---

## [Decision Letter · Decision Letter 1]

22 Nov 2021

PONE-D-21-06570R1Adrenal suppression in patients with chronic obstructive pulmonary disease treated with glucocorticoids: Role of specific glucocorticoid receptor polymorphismsPLOS ONE

Dear Dr. Pradeesh Sivapalan,

Thank you for submitting your manuscript to PLOS ONE. After careful consideration, we feel that it has merit but does not fully meet PLOS ONE’s publication criteria as it currently stands. Therefore, we invite you to submit a revised version of the manuscript that addresses the points raised during the review process.

ACADEMIC EDITOR: the manuscript needs an additional revision following the comments by the reviewer 2.

We look forward to receiving your revised manuscript.

Kind regards,

Fabio Luigi Massimo Ricciardolo

Academic Editor

PLOS ONE

Journal Requirements:

Additional Editor Comments (if provided):

The manuscript has been significantly improved but it needs an additional revision following suggestions by one reviewer.

Reviewers' comments:

Reviewer's Responses to Questions

**Comments to the Author**

1. If the authors have adequately addressed your comments raised in a previous round of review and you feel that this manuscript is now acceptable for publication, you may indicate that here to bypass the “Comments to the Author” section, enter your conflict of interest statement in the “Confidential to Editor” section, and submit your "Accept" recommendation.

Reviewer #1: All comments have been addressed

Reviewer #2: All comments have been addressed

2. Is the manuscript technically sound, and do the data support the conclusions?

Reviewer #1: Yes

Reviewer #2: Yes

3. Has the statistical analysis been performed appropriately and rigorously? 

Reviewer #1: Yes

Reviewer #2: Yes

4. Have the authors made all data underlying the findings in their manuscript fully available?

Reviewer #1: Yes

Reviewer #2: Yes

5. Is the manuscript presented in an intelligible fashion and written in standard English?

Reviewer #1: Yes

Reviewer #2: Yes

6. Review Comments to the Author

Reviewer #1: The authors accurately reviewed the manuscript and is now considered to be suitable for publication in Plos One.

Reviewer #2: Sivapalan and collaborators have significantly improved their manuscript over the previous version. However, this version still had pitfalls and unclear passages, albeit of minor importance.

Why did the number of enrolled patients go from 77 to 78? In the description of the numerosity calculation, the authors wrote: "16 and 48 patients were included in each of the two groups (i.e., 64 patients in total). Assume that 20% of the recruited patients would not be placed in either of the two groups, additional 13 patients were needed, resulting in recruitment of 78 patients to the study". However, adding 13 to 64, the result is 77, as in the previous version of the manuscript, and not 78. Furthermore, a posteriori, it would have been helpful to predict the non-usability of data from "mixed genotype" subjects to recruit more individuals.

Had the authors designed the used primers? If so, how were primers designed? Otherwise, what is their source?

In the text, the descriptions of tables 3 and 5 are missing. Furthermore, given the small population, it would have been interesting to evaluate the characteristics of the "mixed genotype" population. Would intermediate features be conceivable, like wild-type, or would one component prevail?

7. PLOS authors have the option to publish the peer review history of their article (what does this mean?). If published, this will include your full peer review and any attached files.

Reviewer #1: No

Reviewer #2: No

---

## [Author Response · Author response to Decision Letter 1]

10 Dec 2021

Dr. Emily Chenette

Editor-in-Chief

PLOS ONE

Manuscript reference number: PONE-D-21-06570

10 December 2021

Dear Dr. Chenette,

Thank you for the invitation to submit another revised version of our paper: 

PONE-D-21-06570 ‘Adrenal suppression in patients with chronic obstructive pulmonary disease treated with glucocorticoids: role of specific glucocorticoid receptor polymorphisms. 

We would like to thank the reviewers and editors for their valuable comments. We have revised the paper once more according to the comments by Reviewer #2, and we believe it has been substantially improved. Below are our point-by-point responses to the comments made by the reviewers. The reviewers’ comments are listed ‘C, CA, CB, etc.’ and our responses are marked ‘R_C, R_CA, R_CB etc.’ 

Pradeesh Sivapalan MD PhD, Stina Borresen MD PhD, Marianne Klose MD PhD and Professor Ulla Feldt Rasmussen & Professor Jens-Ulrik Jensen MD PhD 

Reviewer #1

C: The authors accurately reviewed the manuscript and is now considered to be suitable for publication in Plos One.

R_C: Thank you, we are pleased to hear so.

Reviewer #2

This reviewer’s comment has been split up in sections and answered accordingly.

C-A: Sivapalan and collaborators have significantly improved their manuscript over the previous version. However, this version still had pitfalls and unclear passages, albeit of minor importance.

R_C-A: Thank you for this comment. We have made substantial changes to clarify several passages, as seen in the new revised manuscript.

C-B: […] Why did the number of enrolled patients go from 77 to 78? In the description of the numerosity calculation, the authors wrote: "16 and 48 patients were included in each of the two groups (i.e., 64 patients in total). Assume that 20% of the recruited patients would not be placed in either of the two groups, additional 13 patients were needed, resulting in recruitment of 78 patients to the study". However, adding 13 to 64, the result is 77, as in the previous version of the manuscript, and not 78.

R_C-B: Thank you for this important comment. Unfortunately, we made a mistake in declaring 77 enrolled patients in the original manuscript. The correct number is 78. Also, it was unclear that the assumption was approximately 20% (i.e. 18%), thus resulting in 14 patients expected not to be placed in either group. It has now been clarified that the 14 additional patients enrolled was a decision based on the approximate assumption, and not exactly equal to the assumed. It now reads as follows:

Page 6, lines 144-146:

“Assuming that approximately 15-20% of the recruited patients would not be placed in either of the two groups, additional 11-16 patients were needed, resulting in a decided recruitment of 78 patients to the study.”

C-C: […] Furthermore, a posteriori, it would have been helpful to predict the non-usability of data from "mixed genotype" subjects to recruit more individuals.

R_C-C: 

Yes, we agree with the reviewer. Unfortunately, there was sparse data on COPD patients, so it was difficult to predict this in our power calculation. We must be aware of this in future studies in this area.

C-D: […] Had the authors designed the used primers? If so, how were primers designed? Otherwise, what is their source?

R_C-D: Yes, the primers were designed by the authors. This, and the design of the primers, has now been clarified in the revised manuscript. It now reads as follows:

Page 5, lines 121-122:

“Primers were designed using Primer-BLAST [16] and ordered through Eurofins[17].”

C-E: […] In the text, the descriptions of tables 3 and 5 are missing.

R_C-E: Thank you for this comment. We have now described table 3 and 5 in the manuscript. Please see below:

Page 9, lines 182-184:

“There was no correlation between accumulated systemic glucocorticoids and 30 min cortisol concentrations in the high-sensitivity haplotype group (p-value, 0.79) or the low-sensitivity plus wild-type haplotype group (p-value, 0.26) (Table 3).”

Page 10, lines 201-204:

“Carriers of the high-sensitivity GR gene haplotype exhibited a steeper decline in stimulated P-cortisol with increased ICS dose compared to patients with the low-sensitivity haplotype plus those with wild-type haplotypes (slope, –1.35 vs. 0.94; p = 0.17; Fig 2 and Table 5).”

C-F: […] Furthermore, given the small population, it would have been interesting to evaluate the characteristics of the "mixed genotype" population. Would intermediate features be conceivable, like wild-type, or would one component prevail?

R_C-F: Thank you for this valuable comment. Characteristics of the mixed genotype population has now been added to Table 1. No significant differences were observed. Mixed type has also been included in Table 3 and Table 5, as well as the total cohort. We agree it would be of great interest to analyse specific haplotypes in the mixed type group as well as the low- and wild groups individually. Power for such analyses, would though require a larger cohort, as multiple haplotypes are present in those groups.

This has now been commented on in the discussion in the manuscript (page 14, lines 300-302):

“Also, it would have been interesting to consider the effect of the different haplotypes present in the mixed genotype and wild type group, but this was not possible with the obtained study sample size.”

---

## [Editor Report · Decision Letter 2]

10 Jan 2022

Adrenal suppression in patients with chronic obstructive pulmonary disease treated with glucocorticoids: Role of specific glucocorticoid receptor polymorphisms

PONE-D-21-06570R2

Dear Dr. Pradeesh Sivapalan,

We’re pleased to inform you that your manuscript has been judged scientifically suitable for publication and will be formally accepted for publication once it meets all outstanding technical requirements.

Kind regards,

Fabio Luigi Massimo Ricciardolo

Academic Editor

PLOS ONE
---

## [Editor Report · Acceptance letter]

24 Jan 2022

PONE-D-21-06570R2 

Adrenal suppression in patients with chronic obstructive pulmonary disease treated with glucocorticoids: Role of specific glucocorticoid receptor polymorphisms 

Dear Dr. Sivapalan:

I'm pleased to inform you that your manuscript has been deemed suitable for publication in PLOS ONE. Congratulations! Your manuscript is now with our production department. 

Kind regards, 

on behalf of

Professor Fabio Luigi Massimo Ricciardolo 

Academic Editor

PLOS ONE